# PARTIAL OPTIMAL TRANSPORT FOR OPEN-SET SEMI-SUPERVISED LEARNING

## ABSTRACT

Semi-supervised learning (SSL) is a machine learning paradigm that leverages both labeled and unlabeled data to improve the performance of learning tasks. However, SSL methods make an assumption that the label spaces of labeled and unlabeled data are identical, which may not hold in open-world applications, where the unlabeled data may contain novel categories that were not present in the labeled training data, essentially outliers. This paper tackles open-set semi-supervised learning (OSSL), where detecting these outliers, or out-of-distribution (OOD) data, is critical. In particular, we model the OOD detection problem in OSSL as a partial optimal transport (POT) problem. With the theory of POT, we devise a mass score function (MSF) to measure the likelihood of a sample being an outlier during training. Then, a novel OOD loss is proposed, which allows to adapt the off-the-shelf SSL methods with POT into OSSL settings in an end-to-end training manner. Furthermore, we conduct extensive experiments on multiple datasets and OSSL configurations, demonstrating that our method consistently achieves superior or competitive results compared to existing approaches.

## 1 INTRODUCTION

Semi-supervised learning (SSL) is a branch of machine learning that leverages both labeled and unlabeled data to improve the performance of learning tasks, as shown in (Lee et al., 2013; Laine & Aila, 2016; Tarvainen & Valpola, 2017; Berthelot et al., 2019; Xie et al., 2020; Sohn et al., 2020). However, a common assumption of SSL is that the label spaces of labeled and unlabeled data are identical, which may not hold in open-world applications. In practice, unlabeled data can contain novel categories that are unseen in the labeled data, i.e., outliers. Existing SSL approaches fall short in supporting such open-set semi-supervised learning (OSSL) scenarios. For example, a common step in SSL is pseudo-labeling (Lee et al., 2013), which often assigns inappropriate pseudo-labels to outliers, thus posing challenges to the generalization over inliers and also the detection of outliers. This leads to a more realistic and practical setting called Open-set Semi-supervised Learning (OSSL) (Yu et al., 2020), which classifies samples of known categories, i.e., inliers, into correct classes, while identifying outliers at test time. However, existing OSSL methods have their limitations. For instance, MTCF (Yu et al., 2020) uses Otsu thresholding to distinguish inliers and outliers, which is not robust if there are a few outliers in the unlabeled data. OpenMatch (Saito et al., 2021) and T2T (Huang et al., 2021) employ a one-vs-all framework, where the error rate is proportional to the number of training categories.

In OSSL, there is a distribution shift (Bickel et al., 2009) problem since the labeled and unlabeled data may come from different distributions. A prevalent way to mitigate the distribution shift is to utilize optimal transport (OT) (Villani, 2008; Courty et al., 2014), a powerful mathematical tool for measuring the discrepancy between distributions. However, the conventional OT assumes that the two distributions have the same total mass and all the mass should be fully transported. Obviously, this assumption can be violated in the context of OSSL, where only a subset of the unlabeled data is aligned with the labeled data, and the rest are outliers. Therefore, a more suitable approach for OSSL is partial optimal transport (POT) (Caffarelli & McCann, 2010), which allows for transporting only a fraction of the mass from one distribution to another. And yet, to the best of our knowledge, POT in OSSL remains unexplored.

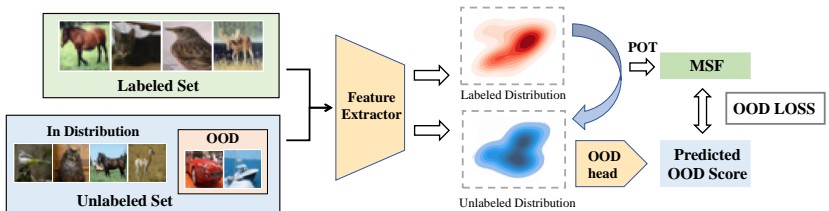

Figure 1: An illustration of OSSL framework with POT. The shared feature extractor receives the labeled and unlabeled data as inputs and produces two kinds of feature representations, forming the corresponding distributions (red area and blue area). The model is iteratively updated under OOD loss, which computes the MSE between the MSF (calculated by POT between two distributions) and the OOD score (predicted by OOD-head).

In this paper, we introduce a novel OSSL approach that utilizes partial optimal transport (POT) to exploit the distribution information of labeled and unlabeled data for outlier detection. In particular, we first formulate the outlier detection problem in OSSL as a POT problem. With the theory of POT, we devise a mass score function (MSF) to measure the likelihood of unlabeled samples being outliers based on the mass transported between the two distributions. MSF is calculated through the optimal transport plan of partial optimal transport, where the MSF for an unlabeled sample is determined by the sum of its mass transport to the labeled distribution. Samples with higher scores are more likely to be identified as ID samples. Then, to train the model in an end-to-end manner, a novel OOD loss is proposed, in which the input is predicted by a designed output layer, called as OOD-head, and the target is the value of MSF produced by POT. Moreover, it allows us to adapt the off-the-shelf SSL methods (e.g. FixMatch) into OSSL settings via multi-task learning. Extensive experiments on multiple datasets and settings demonstrate that our method consistently achieves superior performance on OOD detection for OSSL.

The main contributions of this paper can be summarized as follows:

- We provide the first study that formulates the OOD detection in OSSL as a POT problem.
- With the POT theory, we propose a novel mass score function to measure the likelihood of an unlabeled sample being an OOD sample.
- To train the model in an end-to-end manner, we devise an OOD loss, which allows us to adapt the off-the-shelf SSL method into OSSL settings via multi-task learning.
- Through extensive experiments over multiple OSSL settings, we demonstrate the superiority of our proposed method compared to existing approaches.

## 2 RELATE WORK

### 2.1 OPEN-SET SEMI-SUPERVISED LEARNING (OSSL)

SSL, as a machine learning paradigm, aims to utilize the unlabeled and labeled data to improve the performance of learning (Blum & Mitchell, 1998; Grandvalet & Bengio, 2004; Chapelle & Zien, 2005). Recent breakthroughs like FixMatch (Sohn et al., 2020) and UDA (Xie et al., 2020) combine pseudo-labeling and consistency regularization using data augmentations. However, a common assumption of SSL is that the labeled and unlabeled data share identical label spaces, which often does not hold in real applications, resulting in the study of OSSL. For instance, Chen et al. (2020) proposed **UASD**, which uses softmax prediction probability (MSP) to generate soft targets for the inlier classifier by averaging predictions from multiple temporally ensembled networks. Yu et al. (2020) proposed **MTCF**, which employs a separate open-set recognition (OSR) head for predicting the likelihood of a sample being ID. MTCF utilizes joint optimization, alternating between updating network parameters and estimating anomaly scores for unlabeled data. Guo et al. (2020) proposed **DS3L**, which addresses the OSSL problem via a bi-level optimization task, where inner loss optimizes the parameters of the model, and the outer loss optimizes the weight assigned to unlabeled

samples. Saito et al. (2021) proposed **OpenMatch**, which trains one-vs-all classifiers for each class to identify outliers in an unsupervised way and uses open-set consistency regularization to improve outlier detection in OSSL. Huang et al. (2021) proposed **T2T**, which uses a similar strategy to build one-vs-all classifiers and uses self-supervision and cross-modal match to improve the performance. More recently, Wallin et al. (2023) proposed **SeFOSS**, which uses free-energy score to identify outliers and uses self-supervision to improve the robustness of OSSL.

## 2.2 OUT-OF-DISTRIBUTION (OOD) DETECTION FOR DNNs

OOD detection is to identify whether an input belongs to the same distribution as the training data or not. It holds significant importance in ensuring the reliability and safety of machine learning systems. In existing OSSL research, OOD detection is a pivotal component. A simple OOD detection method for DNNs is the maximum softmax prediction probability (MSP), initially introduced as a baseline by Hendrycks & Gimpel (2016). Subsequently, Liang et al. (2017) enhanced the performance of MSP using temperature scaling and input processing, dubbing their method as ODIN. Also, Liu et al. (2020) proposed a unified framework for OOD detection using an energy score, derived from the log-density ratio between ID and OOD data.

## 2.3 OPTIMAL TRANSPORT (OT)

OT is widely used in machine learning, such as supervised learning (Frogner et al., 2015), generative models (Arjovsky et al., 2017), and graph learning (Kolouri et al., 2021). For example, Courty et al. (2014) proposed a domain adaptation method using regularized optimal transport, which aligns the source and target domains while preserving the inherent structures. Chapel et al. (2020) proposed a PU-learning method based on partial optimal transport. More recently, Lu et al. (2023) applied OT on SCOOD task, to enable the learning of a robust classifier invariant to the input distribution. Xu et al. (2020) and Yang et al. (2023) address open-set domain adaptation problems using POT. Xu et al. (2020) uses the mean cost of transport to control the proportion of POT. Yang et al. (2023) providing an approximate estimation of the transport ratio in POT. Compared to existing POT-based methods, our method stands out by not necessitating prior knowledge about the ratio of POT. These methods underscore the utility of optimal transport in tackling open-world problem.

## 3 PRELIMINARY

In this section, we present essential background information on OT and FixMatch (see A.1), making a basis for subsequent proposed techniques. OT, as discussed in (Villani, 2021), is to seek an optimal transport plan between two measures at a minimal cost, resulting in a metric Wasserstein distance, as explained in (Figalli & Glaudo, 2021), providing a geometric way for aligning probability measures.

### 3.1 DISCRETE OPTIMAL TRANSPORT

Let $\mathcal{X} = \{\mathbf{x}_i\}_{i=1}^n$ and $\mathcal{Y} = \{\mathbf{y}_i\}_{y=1}^m$ be two point clouds representing the source and target samples, respectively. We assume two discrete distributions $(\mathbf{p}, \mathbf{q}) \in \Sigma_n \times \Sigma_m$ over $\mathcal{X}$ and $\mathcal{Y}$. These distributions can be expressed as follows.

$$\mathbf{p} = \sum_{i=1}^n p_i \delta_{\mathbf{x}_i} \qquad \mathbf{q} = \sum_{j=1}^m q_j \delta_{\mathbf{y}_j} \tag{1}$$

where $\Sigma_n$ and $\Sigma_m$ are probability simplex in dimension $n$ and $m$, respectively. The set of all admissible couplings $\Pi(\mathbf{p}, \mathbf{q})$ between the histograms is given by

$$\Pi(\mathbf{p}, \mathbf{q}) = \left\{ \mathbf{T} \in \mathbb{R}_+^{|\mathbf{p}| \times |\mathbf{q}|} | \mathbf{T}\mathbf{1}_{|\mathbf{q}|} = \mathbf{p}, \mathbf{T}^\top \mathbf{1}_{|\mathbf{p}|} = \mathbf{q} \right\} \tag{2}$$

where $\mathbf{T}$ is a coupling matrix with an entry $T_{ij}$ that describes the amount of mass $p_i$ at $\mathbf{x}_i$ transported to the mass $q_j$ at $\mathbf{y}_j$.

Given a transportation cost matrix $\mathbf{C}$, the element $C_{ij}$ represents the cost of transporting one unit of mass from $p_i$ to $q_j$. Optimal transport addresses the problem of transporting $\mathbf{p}$ toward $\mathbf{q}$ with

minimal cost. A coupling matrix achieves the minimal cost is called optimal transport plan and the minimal cost is called optimal transport distance. The OT problem can be described as:

$$\min_{\mathbf{T} \in \Pi(\mathbf{p}, \mathbf{q})} \langle \mathbf{C}, \mathbf{T} \rangle_F = \min_{\mathbf{T} \in \Pi(\mathbf{p}, \mathbf{q})} \sum_{i=1}^{n} \sum_{j=1}^{m} C_{ij} T_{ij} \tag{3}$$

## 3.2 REGULARIZED OPTIMAL TRANSPORT

The discrete optimal transport formulation, in essence, is a convex optimization problem, more precisely, a linear programming problem. Unfortunately, this linear programming problem suffers from a cubic computing complexity. To mitigate this, one way is to leverage entropic regularization, as outlined in (Cuturi, 2013), which is formulated as:

$$\min_{\mathbf{T} \in \Pi(\mathbf{p}, \mathbf{q})} \langle \mathbf{C}, \mathbf{T} \rangle_F - \varepsilon H(\mathbf{T}) \quad where \quad H(\mathbf{T}) := -\sum_{i,j} T_{ij} \big( log(T_{ij}) - 1 \big) \tag{4}$$

where $\varepsilon > 0$ is the regularization coefficient, and $H(T)$ is the entropic regularization term. According to (Benamou et al., 2015), we can recast the regularized OT problem in the language of Kullback-Leibler projections:

$$\min_{\mathbf{T} \in \Pi(\mathbf{p}, \mathbf{q})} \langle \mathbf{C}, \mathbf{T} \rangle_F - \varepsilon H(\mathbf{T}) = \varepsilon \min_{\mathbf{T} \in \Pi(\mathbf{p}, \mathbf{q})} KL(\mathbf{T} | \xi) \quad where \quad \xi = e^{-\frac{\mathbf{C}}{\varepsilon}} \tag{5}$$

The regularized OT is essentially a special case where the feasible set $\mathcal{C}$ is an intersection of two affine subspaces $\mathcal{C}_1, \mathcal{C}_2$:

$$\mathcal{C}_1 = \left\{ \mathbf{T} \in \mathbb{R}_+^{|\mathbf{p}| \times |\mathbf{q}|}; \ \mathbf{T} \mathbf{1}_{|\mathbf{q}|} = \mathbf{p} \right\} \qquad \mathcal{C}_2 = \left\{ \mathbf{T} \in \mathbb{R}_+^{|\mathbf{p}| \times |\mathbf{q}|}; \ \mathbf{T}^\top \mathbf{1}_{|\mathbf{p}|} = \mathbf{q} \right\} \tag{6}$$

This problem can be efficiently solved using Bregman Projection (Benamou et al., 2015).

## 4 METHODS

### 4.1 PROBLEM SETTING

In OSSL, the labeled dataset consists exclusively of in-distribution (ID) samples, while the unlabeled dataset and the test dataset include a mixture of ID and OOD samples. The goal of OSSL is to obtain a model that can correctly classify ID samples into their corresponding classes and distinguish OOD samples at test time. Our proposal materializes the OT theory into SSL framework with distinguishing whether a sample is from the in-distribution to eliminate the negative effect of OOD samples during training. In each training iteration, SSL methods typically sample $n$ labeled samples and $m$ unlabeled samples. The features of these $d$-dimensional samples can be extracted by a feature extractor, resulting in feature representations $\mathbf{L} \in \mathbb{R}^{n \times d}$ and $\mathbf{U} \in \mathbb{R}^{m \times d}$, with corresponding probability distributions $\mathcal{L}$ and $\mathcal{U}$, respectively.

### 4.2 MODELING OOD DETECTION AS POT PROBLEM

OT as a widely-used technique, provides a way to quantify the discrepancy between probability distributions. However, with the obtained population-wise metric, i.e. the distribution discrepancy, it is prohibitive to utilize OT in a direct way to discern individual OOD sample in this setting. Therefore, a natural question arises: how can we utilize OT between labeled distribution $\mathcal{L}$ and unlabeled distribution $\mathcal{U}$ to identify OOD samples in $\mathcal{U}$?

The conventional OT assumes that the two distributions have the same total probability mass, i.e., $\|\mathcal{L}\|_1 = \|\mathcal{U}\|_1$, and that all the mass must be transported. Although this assumption can be well applied to the task of identical distributions or domain adaptation, it becomes problematic when dealing with scenarios such as OSSL, where only a portion of samples in $\mathcal{U}$ share an identical

distribution with $\mathcal{L}$. A straightforward way is to transport the mass of ID samples between $\mathcal{U}$ and $\mathcal{L}$, while keeping the mass of OOD samples in $\mathcal{U}$ stayed. This emerges the concept of POT, which allows to transport a portion of the mass, paving the road for revealing OOD samples.

The problem of POT focuses on transporting only a fraction $0 \leq s \leq min(\|\mathcal{L}\|_1, \|\mathcal{U}\|_1)$ of the mass while minimizing the associated cost. In this case, the set of admissible couplings is defined as:

$$\Pi^s(\mathcal{L}, \mathcal{U}) = \left\{ \mathbf{T} \in \mathbb{R}_+^{|\mathcal{L}| \times |\mathcal{U}|} | \ \mathbf{T}\mathbf{1}_{|\mathcal{U}|} \leq \mathcal{L}, \mathbf{T}^\top \mathbf{1}_{|\mathcal{L}|} \leq \mathcal{U}, \mathbf{1}_{|\mathcal{L}|}^\top \mathbf{T}\mathbf{1}_{|\mathcal{U}|} = s \right\} \tag{7}$$

Hereby, we propose a novel method that uses POT to detect outliers in OSSL settings. To model outlier detection with POT, we first define the labeled distribution $\mathcal{L}$ and K-fold unlabeled distribution $\mathcal{U}$ as follows.

$$\mathcal{L} = \sum_{i=1}^n \frac{1}{n} \delta_{\mathbf{l}_i} \qquad \mathcal{U} = \sum_{i=1}^m \frac{k}{m} \delta_{\mathbf{u}_i} \ (k > 1) \tag{8}$$

where $\mathbf{l}_i$ and $\mathbf{u}_i$ represent the feature of the $i$-th sample in labeled and K-fold unlabeled distributions, respectively. The introduced parameter $k$ denotes the redundant mass, which enables transporting $k$-fold mass from each sample in $\mathcal{U}$. In this way, we represent the resulting problem as a POT problem with the following feasible set.

$$\Pi(\mathcal{L}, \mathcal{U}) = \{ \mathbf{T} \in \mathbb{R}_+^{|\mathcal{L}| \times |\mathcal{U}|} | \ \mathbf{T}\mathbf{1}_{|\mathcal{U}|} = \mathcal{L}, \mathbf{T}^\top \mathbf{1}_{|\mathcal{L}|} \leq \mathcal{U} \} \tag{9}$$

As aforementioned, POT is also a linear programming problem with inequality constraints. To accelerate the calculation, we employ entropy regularization for solving POT. Analogous to Equation 4, the set of feasible solutions $\mathcal{C}$ of regularized POT is an intersection of two convex subspaces, i.e., $\mathcal{C}_1 \cap \mathcal{C}_2$, where

$$\mathcal{C}_1 = \{ \mathbf{T}\mathbf{1}_{|\mathcal{U}|} = \mathcal{L} \} \qquad \mathcal{C}_2 = \{ \mathbf{T}^\top \mathbf{1}_{|\mathcal{L}|} \leq \mathcal{U} \} \tag{10}$$

Equation 9 constitutes a special case of POT, characterized by an inequality constraint and an equality constraint, which is explicitly simpler than the formulation of a general POT problem. Following with previous works (Dykstra, 1983; Bauschke & Lewis, 2000) for solving POT, we conclude our computing framework as Algorithm 1.

---

**Algorithm 1** POT

---

**Require:** labeled distribution $\mathcal{L}$, K-fold unlabeled distribution $\mathcal{U}$, cost matrix $\mathbf{C}$, regularization parameter $\varepsilon$, and error $\delta$.
    **Constrains**: $\mathcal{C}_1 = \{ \mathbf{T}\mathbf{1}_{|\mathcal{U}|} = \mathcal{L} \} \qquad \mathcal{C}_2 = \{ \mathbf{T}^\top \mathbf{1}_{|\mathcal{L}|} \leq \mathcal{U} \}$
    **Initialize**: $\mathbf{T}_0 = e^{-\frac{\mathbf{C}}{\varepsilon}}, q^{(0)} = q^{(-1)} = \mathbf{1}, \mathcal{C}_{n+2} = \mathcal{C}_n$
    **while** $err < \delta$ **do**
        $\mathbf{T}^{(n)} = \underset{\mathbf{T}^{(n)} \in \mathcal{C}_n}{\arg\min} \mathbf{KL}(\mathbf{T}^{(n)} | \mathbf{T}^{(n-1)} \odot q^{(n-2)})$
        $q^{(n)} = q^{(n-2)} \odot \frac{\mathbf{T}^{(n-1)}}{\mathbf{T}^{(n)}}$
        $err = \left\| \mathbf{T}^{(n)} - \mathbf{T}^{(n-1)} \right\|$
    **end while**
    **return** $\mathbf{T}$

---

### 4.3 WHY POT WORKS FOR OOD DETECTION: MASS SCORE FUNCTION

In Section 4.2, we have formulated POT between probability distributions defined over labeled and unlabeled datasets. This formulation urges us to design an effective score function to measure the likelihood of a sample being an OOD sample. To address this, we propose a novel mass score function (MSF) based on the optimal transport plan of POT. MSF represents the sum of mass transported from an unlabeled sample to labeled samples, serving as in indicator for OOD samples.

Next, we describe the mechanism of MSF and explain why POT is effective for OOD detection. The key idea is to leverage the discrepancy between labeled distribution $\mathcal{L}$ and K-fold unlabeled

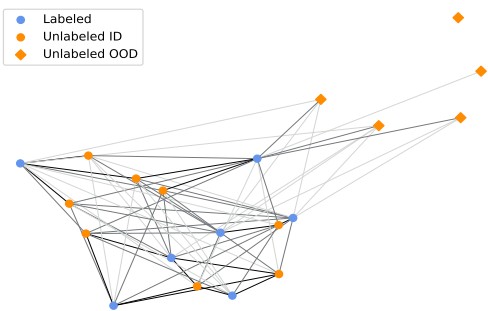

Figure 2: An illustration of mass score function (MSF). The blue dots represent the labeled ID samples (i.e. $\mathcal{L}$), while the orange dots (i.e. $\mathcal{U}$) represent the unlabeled samples which encompass both ID and OOD samples. The circular dots denote unlabeled ID samples (aligned with $\mathcal{L}$) and the diamond-shaped dots denote unlabeled OOD samples (not aligned with $\mathcal{L}$). The depth of the lines between two dots indicates the amount of mass transported from the unlabeled samples to the labeled samples. A darker line indicates a larger amount of mass being transported, and vice versa.

distribution $\mathcal{U}$ to find the part in $\mathcal{U}$ that aligns with $\mathcal{L}$. As presented in Equation 8, we treat $\mathcal{L}$ as a reference distribution, in which samples have the identical weight (i.e., the same mass transported), assured by equality constraints. In contrast, samples in $\mathcal{U}$ may have different mass transported, falling within the range of $[0, \frac{k}{m}]$, as assured by inequality constraints.

The cost of transporting unit mass between ID samples (including labeled samples and ID samples in unlabeled dataset) is small. According to the principle of POT, only a fraction of the mass in the unlabeled distribution can be transported. To minimize the overall transport cost, ID samples in unlabeled dataset obtain larger transport mass. This characteristic aids in identifying OOD samples based on the transport mass of each individual sample.

As illustrated in Figure 2, lines between the labeled samples and the unlabeled ID samples are dense, while lines between the labeled samples and the unlabeled OOD samples are sparse, which indicates the mass of ID samples from the K-fold unlabeled distribution transported to the labeled distribution is much larger than that of OOD samples. Therefore, we can utilize the transport mass as a reliable OOD score, where a sample with smaller value of mass score function tends to be an OOD sample.

To derive the mass score function, we first adopt cosine distance as the transport cost between labeled and unlabeled samples. Then, we use Algorithm 1 to obtain the optimal transport plan. Finally, we calculate MSF and scale it to the range $[0, 1]$ for training. The procedure is detailed in Algorithm 2.

---

**Algorithm 2** POD: POT for OOD Detection

---

**Require:** Feature of labeled and K-fold unlabeled distribution $\mathbf{L}$ and $\mathbf{U}$, entropy regularization coefficient $\varepsilon$, and mass redundancy parameter $k$.
 **Distribution**: $\mathcal{L} = \sum_{i=1}^{n} \frac{1}{n} \delta_{\mathbf{l}_i}$ and $\mathcal{U} = \sum_{i=1}^{m} \frac{k}{m} \delta_{\mathbf{u}_i}$
 **Cost**: calculate cosine distance matrix $\mathbf{M} = Cosine(\mathbf{L}, \mathbf{U})$
 **OT Plan**: calculate optimal transport plan using algorithm1: $\mathbf{T} = POT(\mathcal{L}, \mathcal{U}, \mathbf{M}, \varepsilon)$
 **OOD Score**: calculate optimal transport mass as OOD score for $\mathcal{U}$: $Score_{\mathcal{U}} = \mathbf{T}^{\top} \mathbf{1}_n$
 **return** Scaled $Score_{\mathcal{U}}$

---

### 4.4 OSSL TRAINING WITH POT

In Section 4.3, we present a novel OOD detection method using POT between $\mathcal{L}$ and $\mathcal{U}$. In the training stage, we can use POT to calculate the OOD score for each unlabeled sample. However, in the testing stage, challenges arise in using POT on the test data, since there are no corresponding labeled data for reference. To address this problem, we devise the OOD-head, an additional output layer, to retain the capability of detecting OOD samples supported by POT.

In particular, we denote the score produced by the OOD-head as $Pred_{\mathcal{U}}$ and the value of MSF predicted by POT as $Score_{\mathcal{U}}$. At the training stage, the OOD-head is expected to produce scores

close to the values of MSF for unlabeled samples. To this end, we resort to the following mean square error (MSE) loss.

$$L_{OOD} = \text{MSE}(Pred_{\mathcal{U}}, Score_{\mathcal{U}}) \tag{11}$$

Indeed, the OSSL task can be viewed as a combination of a semi-supervised classification task and an OOD detection task. We therefore formulate the OSSL problem as a multi-task learning problem, which is implemented by only using a simple multi-layer feedforward neural network (Caruana, 1997). More, the two tasks share the same feature extractor and use different prediction heads.

In addition, we outline two strategies that we have implemented to further enhance the performance of our OOD-head. Firstly, we utilize both labeled and unlabeled samples for OOD-head training, as opposed to solely relying on unlabeled samples. Leveraging supervision information from ID labeled samples, we therefore set the OOD score of each labeled sample as 1. Second, we use the same data augmentation strategy with FixMatch for the OOD Detection loss $L_{OOD}$ to improve the generalization capability of OOD-head. We use POT to obtain the MSF for unlabeled samples, based on the features of weakly-augmented versions of unlabeled images. Further, we use the OOD loss to optimize the predicted OOD score for strongly-augmented versions of unlabeled samples. The overall computing framework is described in Algorithm 3.

---

**Algorithm 3** OSSL Computing Framework

---

**Require:** Feature extractor $f$, prediction head $g$, and OOD head $h$. Weight of FixMatch loss $\lambda_u$, weight of OOD loss $\lambda_{OOD}$. Weak data augmentation $\alpha$ and strong data augmentation $\mathcal{A}$. Entropy regularization parameter $\varepsilon$, and mass redundancy parameter $k$.
  Sample a batch of labeled data $D_L$ and a batch of unlabeled data $D_U$
  **Feature Extractor:**
  Get feature of the labeled batch $\mathbf{L} = f(D_L) \in \mathbb{R}^{n \times d}$ and the feature of weakly augmented unlabeled batch $\mathbf{U} = f(\alpha(D_U)) \in \mathbb{R}^{m \times d}$
  **OOD Score:**
  OOD score of unlabeled batch using Algorithm2: $Score_{\mathcal{U}} = \text{POD}(\mathbf{L}, \mathbf{U}, \varepsilon, k)$
  Set the OOD score of labeled samples $Score_{\mathcal{L}} = \mathbf{1}_n$
  $Score = \text{CONCAT}(Score_{\mathcal{U}}, Score_{\mathcal{L}})$
  **Predict OOD Score:**
  The score of labeled batch $Pred_{\mathcal{L}} = h(f(D_L))$
  The score of strongly augmented unlabeled batch $Pred_{\mathcal{U}} = h(f(\mathcal{A}(D_U)))$
  $Pred = \text{CONCAT}(Pred_{\mathcal{U}}, Pred_{\mathcal{L}})$
  **Calculate Loss:**
  calculate $L_x$ using (12), $L_u$ using (14) detailed in Appendix, and $L_{OOD}$ using (11)
  loss = $L_x + \lambda_u L_u + \lambda_{OOD} L_{OOD}$

---

## 5 EXPERIMENTAL RESULTS

**Baselines**. Regarding OSSL baselines, we evaluate our approach, i.e., POT, against existing methods such as MTCF, T2T, and OpenMatch. Additionally, we consider FixMatch as a baseline, although it does not incorporate outlier detection during training. To assess the capability of FixMatch for OOD detection, we employ MSP as score function during testing.

**Evaluation**. For evaluation, we operate under the common OSSL assumption that the test set comprises both known (inlier) and unknown (outlier) classes. To test the performance concerning the known classes, we employ classification accuracy. In order to assess the model's ability in distinguishing between inliers and outliers, we use AUROC, the standard evaluation metric for OOD detection. We report the results averaged over three runs and their standard deviations.

### 5.1 CIFAR10 AND CIFAR100

We evaluate the performance of POT in comparison to baselines on the widely used benchmark datasets for SSL, namely CIFAR10, and CIFAR100. In these experiments, we adapt a randomly initialized Wide ResNet-28-2 (Zagoruyko & Komodakis, 2016) with 1.5M parameters, in consistency with existing works. We follow the approach established by OpenMatch. In particular, for

Table 1: Closed-set accuracy and AUROC on CIFAR10 dataset

| No. of Labeled | 50 | | 100 | | 400 | |
| --- | --- | --- | --- | --- | --- | --- |
| Evaluation | Acc | AUROC | Acc | AUROC | Acc | AUROC |
| FixMatch | $91.7_{\pm 1.1}$ | $37.7_{\pm 0.6}$ | $92.9_{\pm 0.7}$ | $39.8_{\pm 0.5}$ | $93.4_{\pm 0.3}$ | $40.9_{\pm 0.6}$ |
| MTCF | $79.7_{\pm 0.9}$ | $96.6_{\pm 0.5}$ | $86.3_{\pm 0.9}$ | $98.2_{\pm 0.3}$ | $91.0_{\pm 0.5}$ | $98.9_{\pm 0.1}$ |
| T2T | $88.2_{\pm 0.7}$ | $75.5_{\pm 0.5}$ | $89.0_{\pm 1.0}$ | $77.2_{\pm 0.2}$ | $90.3_{\pm 0.5}$ | $82.3_{\pm 0.2}$ |
| OpenMatch | $89.6_{\pm 0.9}$ | $99.3_{\pm 0.3}$ | $\mathbf{92.9}_{\pm \mathbf{0.5}}$ | $99.7_{\pm 0.2}$ | $\mathbf{94.1}_{\pm \mathbf{0.5}}$ | $99.3_{\pm 0.2}$ |
| POT | $\mathbf{92.3}_{\pm \mathbf{0.3}}$ | $\mathbf{99.6}_{\pm \mathbf{0.2}}$ | $92.8_{\pm 0.2}$ | $\mathbf{99.7}_{\pm \mathbf{0.1}}$ | $93.9_{\pm 0.1}$ | $\mathbf{99.6}_{\pm \mathbf{0.1}}$ |

Table 2: Closed-set accuracy and AUROC on CIFAR100 dataset

| No. of Known | 55 | | | | 80 | | | |
| --- | --- | --- | --- | --- | --- | --- | --- | --- |
| No. of Labeled | 50 | | 100 | | 50 | | 100 | |
| Evaluation | Acc | AUROC | Acc | AUROC | Acc | AUROC | Acc | AUROC |
| FixMatch | $78.2_{\pm 0.7}$ | $57.3_{\pm 1.1}$ | $80.8_{\pm 0.6}$ | $56.7_{\pm 1.2}$ | $75.3_{\pm 0.6}$ | $48.7_{\pm 0.9}$ | $78.1_{\pm 0.5}$ | $47.3_{\pm 0.7}$ |
| MTCF | $66.5_{\pm 1.2}$ | $81.2_{\pm 3.4}$ | $72.1_{\pm 0.5}$ | $80.7_{\pm 4.6}$ | $59.9_{\pm 0.8}$ | $79.4_{\pm 2.5}$ | $66.4_{\pm 0.3}$ | $73.2_{\pm 3.5}$ |
| T2T | $72.2_{\pm 1.4}$ | $60.4_{\pm 1.6}$ | $73.1_{\pm 0.8}$ | $59.8_{\pm 1.4}$ | $63.5_{\pm 1.2}$ | $55.0_{\pm 1.8}$ | $66.8_{\pm 0.7}$ | $55.4_{\pm 1.5}$ |
| OpenMatch | $72.3_{\pm 0.4}$ | $87.0_{\pm 1.1}$ | $75.9_{\pm 0.6}$ | $86.5_{\pm 2.1}$ | $66.6_{\pm 0.2}$ | $86.2_{\pm 0.6}$ | $70.5_{\pm 0.3}$ | $86.8_{\pm 1.4}$ |
| POT | $\mathbf{77.6}_{\pm \mathbf{0.2}}$ | $\mathbf{87.5}_{\pm \mathbf{0.1}}$ | $\mathbf{80.2}_{\pm \mathbf{0.4}}$ | $\mathbf{88.0}_{\pm \mathbf{0.4}}$ | $\mathbf{73.9}_{\pm \mathbf{0.1}}$ | $\mathbf{86.8}_{\pm \mathbf{0.4}}$ | $\mathbf{77.1}_{\pm \mathbf{0.3}}$ | $\mathbf{88.3}_{\pm \mathbf{0.7}}$ |

CIFAR10, we divide it into 6 known classes and 4 unknown classes. For CIFAR100, we consider two settings: one with 80 known classes and 20 unknown classes, and another with 55 known classes and 45 unknown classes, organized according to superclasses. Note that we use an identical set of hyper-parameters in all experiments. $\varepsilon$ is set to 0.05, $k$ is set to 2, and $\lambda_{OOD}$ is set to 0.01. A complete list of hyper-parameters is reported in the appendix.

**Comparison to FixMatch.** FixMatch uses a pseudo-label confidence threshold to improve the quality of pseudo-labels in SSL settings. This approach can also be utilized to detect outliers in OSSL settings. As shown in Tables 1 and 2, FixMatch achieves good closed-set accuracy in almost all cases. However, when it comes to the AUROC metric, FixMatch falls short. We delve into a detailed analysis of this phenomenon (see A.2).

**Comparison to OSSL methods.** Tables 1 and 2 show the accuracy for inliers and AUROC values of CIFAR10 and CIFAR100 datasets, respectively. On CIFAR10 dataset, our method stands out by achieving the best results, even with a limited number of labeled samples. This applies for both inlier accuracy and AUROC. Also, our method demonstrates competitive performance compared to SOTA methods when a larger number of labeled samples are used. On CIFAR100 dataset, POT achieves the SOTA performance in all cases. Remarkably, our method significantly improves the classification accuracy compared to OpenMatch. OpenMatch uses one-vs-all framework, training $b$ classifiers to detect OOD samples. However, it is important to note that the cumulative effect of classification errors may lead to a decline in OpenMatch's performance on CIFAR100 dataset. The decline happens due to the usage of multiple classifiers, where the overall error rate becomes 1 minus the product of each individual classifier's error rate (represented by $a$), which is expressed as $1 - (1 - a)^b$, ultimately degrading the performance.

## 5.2 IMAGENET-30

Table 3: Closed-set accuracy and AUROC on Imagenet-30 dataset

| | MTCF | T2T | OpenMatch | POT |
| --- | --- | --- | --- | --- |
| Acc | $86.4_{\pm 0.7}$ | $87.8_{\pm 0.9}$ | $89.6_{\pm 1.0}$ | $\mathbf{90.3}_{\pm \mathbf{0.3}}$ |
| AUROC | $93.8_{\pm 0.8}$ | $55.7_{\pm 10.8}$ | $96.4_{\pm 0.7}$ | $\mathbf{97.4}_{\pm \mathbf{0.4}}$ |

We evaluate POT on ImageNet dataset to observe its performance in more complex and challenging settings. In this assessment, we follow the same experimental configurations as OpenMatch. In particular, we use ImageNet-30 (Hendrycks & Gimpel, 2016), a subset of ImageNet containing 30

classes. We select the first 20 classes in alphabetical order as known classes, where the remaining 10 classes are designated as unknown classes. We employ the ResNet-18 (He et al., 2016) as backbone network. The hyper-parameters are consistent with those employed in CIFAR10 and CIFAR100 experiments. Notably, as shown in Table 3, POT achieves the SOTA performance on the Imagenet30 dataset in terms of inlier accuracy and AUROC.

## 6 ABLATION STUDY

In this section, we study the effect of key components for the performance, including the redundant mass parameter $k$, regularization coefficient $\varepsilon$, and training weight $\lambda_{OOD}$, to demonstrate the robustness of our method.

Table 4: Ablation study on parameter $k$ and regularization coefficient $\varepsilon$ with multiple combinations.

| $k$ | 1.5 | | 2 | | 2.5 | |
|---|---|---|---|---|---|---|
| $\varepsilon$ | Acc | AUROC | Acc | AUROC | Acc | AUROC |
| 0.01 | 92.0 | 98.1 | 92.6 | 98.7 | 91.1 | 99.6 |
| 0.05 | 92.3 | 98.8 | 92.3 | 99.6 | 90.1 | 99.0 |
| 0.1 | 92.1 | 99.5 | 92.6 | 99.1 | 92.3 | 99.6 |

Table 4 shows the performance of the model under 9 different configurations of partial optimal transport parameters. Notably, we observe that when the unlabeled dataset in the CIFAR10 dataset contains 40 percent of the OOD samples, the model achieves excellent results with $k$ ranging from 1.5 to 2.5. This observation underscores the validity of our method, particularly the strategy of generating the redundant mass for unlabeled distributions.

Further, we examine the effect of the entropy regularization parameter on the model. We find that the model produces greater stability with larger regularization parameters. In addition, based on the results presented in Table 4, the model consistently outperforms most parameter combinations, demonstrating the strong generalization capabilities of our proposed method.

Table 5: Ablation study on different weight of OOD loss

| No. of Labeled | 50 | | 100 | | 400 | |
|---|---|---|---|---|---|---|
| $\lambda_{OOD}$ | Acc | AUROC | Acc | AUROC | Acc | AUROC |
| 1 | 92.2 | 87.7 | 92.7 | 97.6 | 93.8 | 89.9 |
| 0.1 | 92.4 | 99.2 | 92.6 | 98.4 | 93.8 | 99.6 |
| 0.01 | 92.3 | 99.6 | 92.7 | 99.7 | 93.9 | 99.6 |
| 0.001 | 91.5 | 97.5 | 93.0 | 99.0 | 93.7 | 99.6 |

The impact of varying the weight assigned to the OOD loss is demonstrated in Table 5. The result shows that our proposed approach yields superior performance when tuning the OOD loss weights to align with the order of magnitude of the supervised loss (e.g., weights of 0.1 or 0.01). This suggests that the OOD loss should have a balanced weight with the supervised loss, neither dominating it nor being dominated by it. In practice, since the supervised loss typically carries a smaller magnitude, it suggests to use small weights for the OOD loss when applying our method.

## 7 CONCLUSION

In this work, we study the problem of OSSL from the perspective of POT. In particular, we formulate the OOD detection in OSSL as a POT problem. With the POT theory, we propose a novel mass score function based on transport mass to detect outliers. To train the model in an end-to-end manner, we devise an OOD loss, adapting the off-the-shelf SSL method into OSSL settings via multi-task learning. Our experiments on different datasets and OSSL configurations show that our technique achieves superior results compared to existing methods, all while maintaining simplicity and efficiency. Our technique provides a new perspective and a useful tool for OSSL research, with the hope of inspiring further investigation this challenging and important problem.

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

## A   APPENDIX

### A.1   FIXMATCH

FixMatch is a commonly used semi-supervised learning method. The loss of FixMatch is composed of supervised loss $l_x$ for labeled samples and unsupervised loss $l_u$ for unlabeled samples. FixMatch computes a model's predicted class distribution given a weakly-augmented version of an unlabeled image, and then generates a pseudo-label based on the highest probability class. The method then enforces the cross-entropy loss against the model's output for a strongly-augmented version of the same image, using the pseudo-label as the target.

When the batchsize is $B$, the supervised loss $L_x$ is:

$$L_x = \frac{1}{B}\sum_{b=1}^{B}H(p_b, \alpha(x_b)) \tag{12}$$

where $H$ is the cross-entropy function, $p_b$ is the label for the $b_{th}$ labeled sample, $\alpha$ is a weak augmentation function, and $x_b$ is the original image.

Compute prediction after applying weak data augmentation of $u_b$

$$q_b = p_m(y|\alpha(u_b); \theta) \tag{13}$$

where $p_m$ is the model's prediction function, y is the class label, $u_b$ is the $b$-$th$ unlabeled sample, and $\theta$ represents model parameters.

The unsupervised loss $L_u$ is:

$$L_u = \frac{1}{\mu B}\sum_{b=1}^{\mu B}H(argmax(p_b), p_m(y|\mathcal{A}(u_b))) \tag{14}$$

where $\mathcal{A}$ is a strong augmentation function, and $argmax(p_b)$ is the pseudo-label generated from the weakly-augmented prediction. The original FixMatch unsupervised loss is coupled with a threshold on the confidence of pseudo labels to improve the quality of pseudo labels. The confidence threshold can also serve as a simple OOD detector. We make detailed analysis in the experiment section. Because we add extra OOD detector, we remove the confidence threshold.

### A.2   ANALYSIS OF FIXMATCH

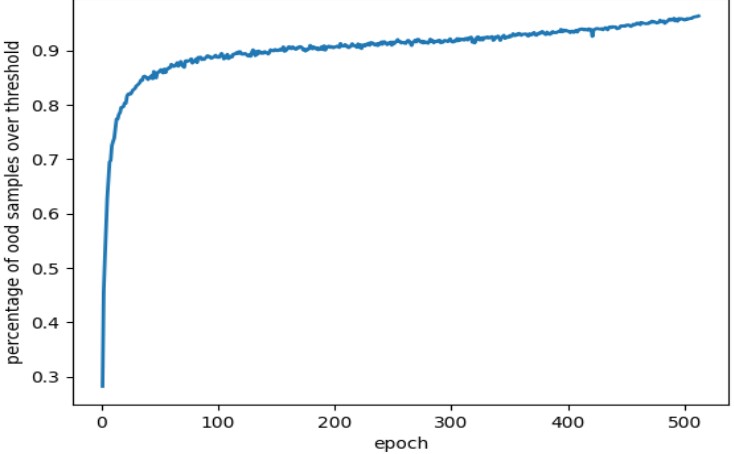

Figure 3: The percentage of OOD sample over FixMatch threshold

FixMatch relies on MSP to detect outliers, whereas softmax-based confidence estimation drawn by a single DNN can be problematic, due to its overconfidence tendencies (Nguyen et al., 2015).

Consequently, it may result in a substantial number of unlabeled samples being erroneously labeled, ultimately degrading the AUROC performance. As Fig. 2 shows, FixMatch assigns high-confidence pseudo-labels to more than 95 percent of the OOD samples, which means they are used for training. This leads to poor performance of FixMatch on the AUROC metric.

## A.3 EXPERIMENTAL DETAILS

Following the same parameter settings oef original paper, we reproduce our baselines using their official codes. For a fair comparison, we unified the parameters of semi-supervised learning as follows.

Each experiment is done with a single NVIDIA A100 GPU.

The batch size $B = 64$.

The relative size of batch-size for unlabeled data $\mu = 2$ in all experiments for fair comparison.

Iterations per epoch are 1024 and the total number of epochs is 512.

Optimizer: SGD with nesterov momentum = 0.9.

Learning rate $\eta = 0.03$ and use cosine annealing learning rate schedules.

## A.4 TRAINING TIME.

OpenMatch employs a two-stage training approach. In the first stage, it selects a batch of labeled and unlabeled samples to identify pseudo-inliers. The second stage involves sampling a batch of pseudo-inliers to mitigate the impact of outliers. On the other hand, our approach utilizes multi-task learning, simplifying the process to a single stage. For the computation complexity of calculating MSF, POT offers an $O(mn)$ complexity, which is lower than the computation cost associated with SSL. Further, when compared to the original FixMatch, we eliminate the need for a confidence threshold. Consequently, the training time for our method closely approximates that of FixMatch. We evaluated the training time of our method, FixMatch, and OpenMatch on CIFAR10 with 400 labeled samples per class. The training time for a single epoch is reported in Table 6.

Table 6: Training time of OSSL methods.

| FixMatch | OpenMatch | POT |
| --- | --- | --- |
| 68s | 138s | 61s |

